# Application of Silver Nanoparticles in Parasite Treatment

**DOI:** 10.3390/pharmaceutics15071783

**Published:** 2023-06-21

**Authors:** Ping Zhang, Jiahao Gong, Yan Jiang, Yunfeng Long, Weiqiang Lei, Xiuge Gao, Dawei Guo

**Affiliations:** 1College of Animal Science and Food Engineering, Jinling Institute of Technology, 99 Hongjing Road, Nanjing 211169, China; zp@jit.edu.cn (P.Z.); leiweiqiang@jit.edu.cn (W.L.); 2Engineering Center of Innovative Veterinary Drugs, MOE Joint International Research Laboratory of Animal Health and Food Safety, College of Veterinary Medicine, Nanjing Agricultural University, 1 Weigang, Nanjing 210095, China; 2022007053@stu.njau.edu.cn (J.G.); vetgao@njau.edu.cn (X.G.); 3Animal, Plant and Food Inspection Center of Nanjing Customs District, 39 Chuangzhi Road, Nanjing 210000, China; jiangyan@customs.gov.cn (Y.J.); xiaomilan126@163.com (Y.L.)

**Keywords:** silver nanomaterials, application, parasitic diseases

## Abstract

Silver nanoparticles (AgNPs) are ultra-small silver particles with a size from 1 to 100 nanometers. Unlike bulk silver, they have unique physical and chemical properties. Numerous studies have shown that AgNPs have beneficial biological effects on various diseases, including antibacterial, anti-inflammatory, antioxidant, antiparasitic, and antiviruses. One of the most well-known applications is in the field of antibacterial applications, where AgNPs have strong abilities to kill multi-drug resistant bacteria, making them a potential candidate as an antibacterial drug. Recently, AgNPs synthesized from plant extracts have exhibited outstanding antiparasitic effects, with a shorter duration of use and enhanced ability to inhibit parasite multiplication compared to traditional antiparasitic drugs. This review summarizes the types, characteristics, and the mechanism of action of AgNPs in anti-parasitism, mainly focusing on their effects in leishmaniasis, flukes, cryptosporidiosis, toxoplasmosis, *Haemonchus*, *Blastocystis hominis*, and *Strongylides*. The aim is to provide a reference for the application of AgNPs in the prevention and control of parasitic diseases.

## 1. Introduction

Parasitic diseases, caused by parasitic worms in the bodies of humans and animals, are mostly zoonotic. Parasites can be divided into three main categories. One is protozoa, such as *Plasmodium* and *Jagandii flagellates*, and this type of parasite is wide-spread. The second is invertebrates, and this type of parasite is the most numerous in terms of number and species; common ones include endoparasitic flatworm, pork tapeworm, Chinese liver flukes, ectoparasitic arthropod pubic lice, head lice, and Culex. The third is vertebrates, and this type of parasite is very rare. Five of the six major tropical disease categories in the joint UNDP/WHO Special Program for Tropical Diseases are parasitic diseases, and 11 of the 17 neglected tropical diseases defined by WHO are parasitic, such as schistosomiasis, encysticercosis, tape-worm/cysticercosis, foodborne trematode, leishmaniasis, etc. [1].

At present, the control of parasitic diseases mainly relies on drugs and vaccine immunization [2]. Though these drugs and vaccines play a great role in the control of parasitic diseases, at the same time, the problems of drug resistance, drug residues, worm strain variation, vaccine side effect, and vaccine safety have also arisen [3,4]. Related studies have shown that anti-parasitical drugs can turn almost all parasites into resistant strains, and the degree of parasite resistance has been increasing and developing from single-drug resistance to multi-drug resistance, which causes the therapeutic effect of anti-parasitic drugs to be insignificant or even useless [5,6]. Even if the drug concentration is increased, the purpose of controlling parasites cannot be achieved. Moreover, excessive drug doses can cause damage to the hosts, as well as create cross-resistance and multi-drug resistance [7]. Therefore, to control parasites, it is not only necessary to analyze the mechanism of drug resistance generation of various parasites and prevent or reverse the occurrence of resistance, but also to require technological innovation to develop new, safer, less toxic anti-parasitic drugs and new drug dosage forms [7] so as to more effectively reduce the transmission of parasitic diseases and mitigate the damage to hosts [8].

In recent years, the rapid development and integration of biotechnology and nanotechnology, as well as their penetration and application in drugs, have greatly contributed to the development of drug discovery [9,10]. The use of nanotechnology in combination with traditional metal inorganic bactericides to develop new antiparasitic drugs is a good choice [11], and the study of metal nanomaterial inhibition and insecticidal mechanism provides a basis for the further development of new parasite inhibitors [12]. As inorganic bactericidal materials, silver ions have a multilocation effect and wide range of pathogens [13,14,15]. However, the application of silver-based materials is greatly limited by the high material cost, unstable chemical properties of free silver ions, and high toxicity of common silver products [16]. Sodium silver materials made by nanotechnology have the advantages of high efficacy in killing pathogens, less resistance to germs, fewer dosages, and chemical stability [17]. The use of AgNP materials for parasite control can provide new substances for the chemical control of clinical parasitic diseases, and the study of the inhibitory effect of AgNPs on parasites and their mechanisms can provide a theoretical basis for the further development of AgNPs parasiticides. This review focuses on the types and characteristics of AgNPs and their application to parasitic diseases such as leishmaniasis, flukes, cryptosporidiosis, toxoplasmosis, *Haemonchus*, *Blastocystis hominis*, and *Strongylides*. We discuss its primary mechanism of action: the disruption of fluidity and integrity in the parasite cell membrane, which leads to increased permeability and loss of intracellular essence, and the release of reactive oxygen species (ROS), which leads to oxidative stress and damage to cellular components [18,19].

## 2. Synthesis and Potential Applications of Silver Nanomaterials

### 2.1. The Properties of Silver Nanomaterials

Nanoparticles are particles with dimensions in the range of 1–100 nm scale in at least one dimension in a three-dimensional space [20]. Materials composed of nanoparticles have many specific properties, such as small size effect, volume effect, interface effect, and macroscopic quantum tunneling effect. Therefore, nanomaterials are known as new materials in the 21st century and are used in the fields of information, biology, medicine, chemical industry, aerospace, energy, national defense, etc. Nanomaterials have broad application prospects and, among which, AgNPs are by far the most abundant commercialized nano-compound in the market used in various areas of daily life. Currently, 435 out of 1814 nanoproducts in 32 countries or regions worldwide contain AgNPs, accounting for 24% of the total [21]. 

AgNP is the abbreviation or common name for silver nanoparticles, which are particles composed of silver atoms, usually in the size range of 1–100 nm. Like bulk silver materials, the surface of AgNPs is oxidized and release free silver ions. However, AgNPs have some characteristics that ordinary materials do not have, such as surface effect, quantum size effect, small size effect, etc. [22]. Due to the small size effect and surface effect of nanoparticles, the release rate of silver ions is significantly higher, and AgNPs can cause direct damage to cell membranes through silver ions, increase the permeability of the cell membrane, and enter many cells, eventually causing apoptosis or necrosis. Due to these properties, the bactericidal effect of AgNPs is significantly higher than that of silver ions. 

In addition, the toxic effects of AgNPs are related to their other characteristics, such as shape, concentration, chemical coating, surface charge, etc. [23]. AgNPs have a wide variety of morphologies, such as spherical, conical, disc, rod, cube, prism, ring, sheet, and triangular prism [24]. Sharp and irregular shapes contribute to the occurrence of physical damage. For example, in the coercion of *Escherichia coli* with triangular, spherical, and rod-shaped AgNPs, the results showed that triangular AgNPs had a stronger antibacterial ability [25], and the coercion of zebrafish embryos with spherical and flaky AgNPs showed that the flaky AgNPs exhibited a stronger toxicity [26]. However, the correlation between particle shape and toxicity is not clear, as it may depend on multiple factors rather than one [23], such as size effect, ionic effect, and surface modifiers that all significantly contribute to the biotoxicity of AgNPs. During the manufacturing process of AgNPs, coatings are often added to their surface to prevent aggregation, which helps increase stability and promotes particle dispersion. There are various types of coatings that can change the morphology of AgNPs and prevent the oxidation of silver ions. This modification has a direct impact on the biotoxicity of AgNPs [27]. It was found that AgNPs modified by citrate and chitosan were more toxic to bacteria than the unmodified ones, probably because these two modifiers accelerated the release of silver ions from AgNPs [28]. The toxicity of AgNPs modified by citrate is less than that of AgNPs modified by polyvinylpyrrolidone and polyethyleneimine or with no coating on the surface [29]. The difference in toxicity between the different coatings may be due to the changes of the surface charge type of the AgNPs caused by coating. The positively charged AgNPs can adsorb directly onto the negatively charged bacterial cell wall and, therefore, exhibit a stronger bactericidal effect than the negatively charged AgNPs [30].

### 2.2. The Synthesis of Silver Nanoparticles

There are many ways to prepare nanomaterials, and the two most basic principles at present are as follows: first, the splitting of large solids into nanoparticles; second, the formation of particles by the aggregation of individual basic atoms and the control of the growth of the particles to maintain them at nanometer size. According to the principle of nanoparticle preparation, the preparation methods of AgNPs can be generally classified into physical synthesis methods [31,32,33,34], chemical synthesis methods [35,36,37,38,39], and biological synthesis methods (Figure 1) [40,41,42,43,44,45]. 

The physical synthesis method includes mechanical grinding, laser ablation, evaporation–condensation, electrical irradiation, gamma irradiation, lithography, arc discharge method, etc. The mechanical grinding method produces local high pressure by high speed collision, thus grinding the metal into a very fine powder, and the size of the nanoparticles depends on the degree of abrasion [46]. The arc discharge method can prepare silver nanoparticles in pure water without any surfactant or stabilizer by an arc-discharge device (Figure 2) [47]. When two silver electrodes form an arc in ionized water, the silver electrodes evaporate and form nanoparticles [32]. The laser ablation method uses a pulsed laser to instantaneously heat a silver block immersed in water or an organic solvent; during the cooling of the plasma, silver particles nucleate and grow, eventually forming nanosilver [48]. Under the evaporation–condensation method, metallic silver is evaporated and condensed into nanoparticles, which are further condensed into atomic clusters or nano-silver particles [32]. Gamma rays can induce a radioactive decomposition of solvents, generating dissolved electrons that can reduce metal ions (such as Ag^+^) in a solution to form nanoparticles [49]. 

Although the physical synthesis method is simple in principle, it is costly to prepare and requires high precision in equipment, which is not suitable for large-scale production [50,51]. 

Chemical synthesis is the most commonly used method to synthesize silver nanoparticles. This method reduces silver ions to elemental silver or silver nanoparticles by electron transfer under certain conditions. Chemical methods can accelerate nanosilver preparation with the help of external energy processes, such as photochemical, electrochemical, microwave-assisted, and acoustic methods [47]. In photochemical methods, light, usually ultraviolet (UV) light, is used to induce the reduction of silver ions to form silver nanoparticles [52]. The photon energy provided by light can excite silver ions and promote the reduction reaction [53] (Figure 3). The electrochemical method involves applying an electromotive force to induce the reduction of silver ions to silver nanoparticles. This is usually conducted in an electrochemical cell with silver electrodes. The applied potential transfers electrons to silver ions, reducing them to silver atoms, which then aggregate into nanoparticles [54]. Microwave-assisted synthesis uses microwave radiation to heat the reaction mixture and promote the reduction of silver ions to nanoparticles [55]. In the context of nanoparticle synthesis, ultrasound can be used to generate cavitation bubbles in liquids. The collapse of these bubbles generates intense localized heat and pressure that can reduce silver ions to nanoparticles [56].

The chemical synthesis method uses reducing reagents to reduce silver ions to silver atoms, which further aggregate into oligomeric clusters to produce metallic colloidal silver particles [36,37]. Commonly used reducing agents include sodium borohydride, hydrazine, ethylene glycol, and N,N-dimethylformamide, etc. [32]. In addition to reducing agents, the chemical synthesis of AgNPs also requires polymers as stabilizers to enhance the stability of AgNPs and avoid its agglomeration, such as polyvinylpyrrolidone, dodecanethiol, and polyvinyl alcohol. However, the chemical synthesis of AgNPs faces a major problem, namely, the toxicity of chemical reagents. Additionally, the selected reducing agents and stabilizers have certain toxic effects on the organism, so the chemically synthesized AgNPs are biotoxic, which also limits their application [57]. 

The biological method includes the plant method and microbial method (Figure 4), which consist of proteins, sugars, and antioxidants derived from organic organisms such as bacteria, fungi, yeast, and plants (tea, seaweed, mustard, etc.) instead of toxin-reducing and stabilizing agents [58,59]; the possible mechanisms of their synthesis are enzymatic and non-enzymatic reductions. 

In the botanical approach, various plant extracts are used to reduce silver ions to silver nanoparticles. Bioactive compounds present in plant extracts, such as phenolic compounds, flavonoids, terpenoids, etc., can act as both reducing agents and stabilizers. The process is usually simple, cost-effective, and environmentally friendly. This approach can utilize a wide variety of plant sources, including tea leaves, seaweed, mustard, etc. [41]. In the microbial method, microorganisms such as bacteria, fungi, and yeast are used to synthesize silver nanoparticles. Microorganisms can produce enzymes that reduce silver ions to elemental silver. During this process, the nanoparticles are usually wrapped in a protein layer that helps stabilize them. Specific strains of bacteria and fungi have been studied for their ability to reduce metal ions and generate nanoparticles [60].

This biosynthesis is characterized by green and environmental protection, uniform and very small particle size, good dispersion, difficulty to precipitate, etc. [61,62], but due to the reduction of particle size, the number of surface atoms increases, which easily leads to the agglomeration phenomenon of nanoparticles [63,64]. 

Because AgNPs tend to self-agglomerate when used alone as an antimicrobial solution, the antimicrobial effect is not fully realized, so AgNPs are often used for loading with other materials [65], such as AgNPs-hydroxyapatite composites [66], Poly (vinyl alcohol)- AgNP (PVA-AgNP) [67], AgNP-TiO_2_ composites [68], Ag/ZnO nanocomposites [69], etc. The composite of AgNPs with other materials increases the compatibility for specific applications extending the unique properties of AgNPs to a broader space. For example, AgNP-TiO_2_ composites have good biocompatibility and high antibacterial activity [70], and Ag/ZnO nanocomposites with silver nanoparticles loaded on ZnO surface showed an inhibitory effect on *Streptococcus* mutans with better antibacterial activity than ZnO nanorods [71].

### 2.3. Potential Applications of AgNPs

Research on AgNPs has shown promising results for in vivo applications. In vivo studies have also demonstrated the antibacterial and anticancer properties of AgNPs. It has been reported that AgNPs could exhibit a bioefficacy on a plant–parasitic nematode against a root–knot nematode on bermuda grass [72] and *Meloidogyne graminicola* [73]. Subsequent studies have explicitly revealed promising results of AgNPs against *Meloidogyne incognita* on eggplant, tomato, and okra [74]. In animal studies, AgNPs have been shown to be effective in treating various infections, including skin, respiratory, urinary tract, and parasitic infections [75]. In addition, AgNPs have been shown to inhibit tumor growth and improve survival in animal models of cancer [76]. 

AgNPs are used in the field of medicine for applications in humans such as drug delivery, cancer therapy, bioimaging, and dental technology. AgNPs are able to get more attention in cancer therapy because of their unique physicochemical properties [77]. The use of metal nanoparticles, compared to conventional anticancer tools, is a new combination of therapeutic drugs and drug carriers with drug candidates, where side effects can be prevented by targeted approaches [78]. In experiments targeting human cervical cancer cells, AgNPs were extracted using *Nepeta deflersiana* (ND), yielding face-centered cubic structures with an average size of 33 nm, by targeting human cervical cancer cells (HeLa) for their anticancer potential by observing the cytotoxic response where the neutral red uptake assay and morphological changes, cytotoxic concentrations on oxidative stress markers, ROS production, and mitochondrial membrane potential parameters responded to cytotoxicity depending on the concentration. Potential mitochondrial membrane and glutathione levels decreased and AgNPs induced apoptosis, demonstrating that ND-AgNPs have an anticancer ability and can be used to treat cervical cancer cells [79]. Nanoparticles also have applications in cellular bioimaging and cell sensing, where they are selected based on their optical effects in order to achieve effective contrast in cellular imaging and other therapies [80]. AgNPs are also widely used in dentistry, where they are incorporated into some dental biomaterials for reducing biofilm formation due to their antibacterial activity, and are incorporated into root canal fillings to reduce *Staphylococcus aureus* and *Streptococcus* mutans [81].

## 3. Antiprotozoal Effect of AgNPs

### 3.1. Leishmaniasis

Leishmaniasis is a protozoan disease caused by the leishmaniasis parasite and transmitted through the bite of sand flies, mainly found in tropical and subtropical regions. There are three main forms of leishmaniasis [82]: First, there is visceral leishmaniasis, also known as black fever, which, if left untreated, causes death in more than 95% of cases. The second is cutaneous leishmaniasis, the most common form, which causes mainly skin lesions such as ulcers in exposed areas of the body. The third is cutaneous mucosal leishmaniasis, which causes partial or total destruction of the mucous membranes of the nose, mouth, and throat. There are some problems in the treatment of leishmaniasis, such as toxicity and drug resistance, which require the development of new drugs to improve treatment.

A previous study has shown that *Teucrium stocksianum* is one of the recommended plants in the green synthesis method of AgNPs, and the anti-leishmanial effect of AgNPs made from the leaves is the best [17]. Other studies have also found that there are other plants that excel in antileishmanial. Additionally, a new plant, *Moringa oleifera*, was also identified as a raw material for the preparation of AgNPs, as this plant is used to prevent malaria, trypanosomiasis, schistosomiasis, and filariasis [83]. Moreover, the researchers found that the average lesion size was reduced and contributed to complete healing in 14 days after the use of AgNPs, which was reduced by half compared to the 28 days needed for standard drugs [83]. At the same time, there is also Iranian research showing that AgNPs made of ginger have the same function [84]. Additionally, the inhibitory ability of different concentrations on the number of promastigotes at 24, 48, and 72 h was observed, and high concentrations of AgNPs could completely inhibit the proliferation of promastigotes at any time period. In an anti-amastigote assay and flow cytometry assay, the participation of AgNPs reduced the number of infected macrophages from a 30% infection rate to 14.75% and increased the number of necrotic and apoptotic cells induced by AgNPs. The number of previable cells was 99.59% and the number of apoptotic and necrotic cells after intervention was 60.18% and 0.53%, respectively [83].

### 3.2. Flukes 

Flukes belong to the class Trematoda of the phylum Platyhelminthes, and their bodies are lobulated or lingual, with oral and ventral suckers. There are more than 30 species of trematode parasites in the human body, and the common ones are *Clonorchis sinensis*, *Fasciolopsis buski*, *Paragonimus westermani*, etc. Flukes affect animal health by reducing milk production and jeopardizing meat health, reducing animal draught power, which in turn increases the chance of animal morbidity and mortality. In infections, most flukes infect the bile ducts of buffalo and water buffalo, while larvae cause intestinal bleeding and adults cause bile duct hypertrophy and hyperplasia, affecting overall health [85]. 

Researchers in India investigated the effects of AgNPs on adult worms in vitro by incubating worms in fetal bovine blood, controlling AgNPs at different concentrations and observing worm motility in 4 h cycles until 16 h later while recording motility without AgNPs intervention. The results showed that the motility of AgNP-treated flukes was reduced compared to the untreated control [86]. Changes in the production of ROS by flukes in the explants were also recorded, showing a concentration-dependent increase in ROS production in AgNP-treated worm cells and an increase in light absorption levels compared to control worms [86]. Other values, such as the estimation of superoxide dismutase (SOD) activity, an oxidative enzyme whose role is to ROS, were significantly reduced, and AgNPs inhibited this enzyme [86]. The effect on DNA fragmentation showed that the treatment of worms with AgNPs could lead to apoptosis, and the level of protein carbonylation was significantly increased after intervention with AgNPs, as measured by the protein carbonylation reaction [86].

### 3.3. Toxoplasmosis 

Toxoplasmosis is an infectious disease caused by *Toxoplasma gondii*, which is parasitic in cells and travels with the bloodstream to reach various parts of the body. *Toxoplasma gondii* can destroy the brain, heart, and fundus of the eyes [87], decrease immunity, and cause various diseases. In an end-host cat, Toxoplasma gondii completes the intestinal stage of development. The worm colonizes the intestinal epithelial cells to develop and proliferate to form oocysts, which are then destroyed and returned to the intestinal lumen and are excreted in the cat’s feces. Oocysts then enter the environment to develop into infective mature oocysts over 2–4 days, posing a threat to other susceptible animals. *Toxoplasma gondii* affects young cows more [88] due to the fact that colostrum and milk are consumed before and after the cow gives birth. Colostrum and milk can be infected with eggs containing larvae [89]. Although most infected individuals are asymptomatic, the disease is considered a significant health concern due to the occurrence of congenital transmission [90]. The severity of *Toxoplasma gondii* disease also depends on the strain itself. The eggs hatch into larvae in the small intestine [91], which then migrate to various tissues, such as the liver, lungs, muscle tissue, and brain, causing damage to these tissues and organs [92]. Symptoms of infection include anorexia, weight loss, and, in severe cases, death, essentially resulting in poor milk and meat quality in cattle, as well as reduced skin quality [93]. Several studies have shown that *Toxoplasma gondii* has developed the ability to control host immune responses and that it can manipulate the host to reduce the amount of parasite suppressor antigens and thus avoid recognition and clearance by the host [94]. 

Clindamycin is the first choice for the treatment of *Toxoplasma gondii*, but the drug has a certain irritating effect on the stomach and intestines. Sulfonamides can inhibit the metabolism of folic acid of the worms and thus inhibit the growth of *Toxoplasma gondii* [95], but the drug can cause a deficiency of folic acid in the diseased animals [96]. Azithromycin is more effective in the treatment of eye diseases caused by *Toxoplasma gondii*, but the efficacy is only for eye diseases. Nanoparticles as drug carriers can reduce the toxicity and improve the bioavailability of parasitic drugs [97]. Existing studies have demonstrated the potential therapeutic ability against parasitic infections [98] of AgNPs, inhibiting the growth of *Toxoplasma gondii* in vitro [99]. In addition to their potential role in biomedicine, AgNPs are now an attractive option to fight parasites, particularly against *Toxoplasma gondii*, as AgNPs can help macrophages fight pathogens [100]. It was found that ROS acts as an important effector molecule that induces cells to eliminate pathogens, as well as signaling molecules that amplify the antimicrobial response by activating the transcription factor nuclear κB (NF-kB), which in turn enables the production of proinflammatory cytokines [101]. AgNPs are able to reduce levels of the antioxidant glutathione in macrophages, which can also be independent of immune regulation accumulates in mitochondria and reduce the formation of adenosine triphosphate, which in turn disrupts the inner mitochondrial membrane, all in a manner that increases ROS production [102], which then leads to the elimination of the parasite. After incubation of adult T. vitulorum worms in 200 mg/L AgNPs for 48 h, light and electron microscopy studies revealed disorganized cuticles damaged subcutaneous blisters, and muscle layers in both male and female worms. Electron microscopy of the treated worms confirmed wrinkled epidermal surfaces and disrupted surface structures. Following the use of AgNPs in the study, nitric oxide levels were recorded in the worms, and levels were significantly increased. Nitric oxide is highly reactive with other concentrated oxidizing molecules such as ROS, producing reactive nitrogen that can attack biological systems, leading to severe irreversible damage to biomolecules and cellular damage. However, the anti-creep efficiency of AgNPs against nematodes in vivo remains to be investigated [103]. Further studies on the in vivo studies and potential value of AgNPs against *Toxoplasma gondii* are needed [104].

### 3.4. Cryptosporidium

Cryptosporidium is a protozoan, which was considered to be an important major cause of enteric parasitic infection, and it is a zoonotic disease that can infect humans, animals, and birds with morbidity and mortality, especially among immune-deficient individuals [105]; so, it is considered a serious public health risk. It is also the most common cause of diarrhea in children worldwide [106] because younger children are more susceptible to Cryptosporidium infection [107,108]. Many birds, including chickens, geese, ducks, and pigeons, are known to be biological reservoirs of Cryptosporidium [109]. The protozoan can be transmitted to humans and animals mainly through the ingestion of contaminated food and drink or through contact with excreta. When humans are infected, Cryptosporidium may cause joint pain and intestinal disorders [110]. Infection with microbial pathogens leads to apoptosis and programmed cell death, and cytochrome C (Cyto) also plays a key role in the process of apoptosis, accelerating it [111]. 

The antiprotozoal activity of the biosynthesized AgNPs on the viability of *Cryptosporidium parvum* (*C. parvum*) oocysts was tested in vitro and in vivo. The results of in vitro experiments showed that the biosynthesized AgNPs had an inhibitory effect on *C. parvum* oocyst viability. Additionally, a significant decrease in *C. parvum* counts was observed after 3, 6, 12, and 24 h of storage for AgNP-treated control *C. parvum* oocysts at concentrations of 500 µg/mL and 1000 µg/mL, and no oocysts were detected after 48 h of storage for those that were AgNP-treated. The results of the in vivo infectivity test showed that for mice infected with AgNP-treated *C. parvum* oocysts, *C. parvum* oocyst counts were significantly lower than that in the control group and showed no infection for AgNPs of 1000 µg/mL for 48 h [109]. Similar results were obtained by Cameron, et al. [12].

Nitazoxanide (NTZ) is the only clinical drug used, but it is of limited use in some patients [105]. The experimental animals selected by the researchers were rats, which were gavaged orally using a solution of AgNPs at a dose of 5 mg/kg/mouse/day NTZ and an oral dose of 100 mg/kg/mouse/day with a mixture of both AgNPs (5 mg/kg) and NTZ (100 mg/kg) that were vortexed for one hour in a closed container beforehand. Cryptosporidium levels were observed by collecting fecal pellets from mice, and the shedding of Cryptosporidium oocysts was the criterion for the effectiveness of the drug. The group treated with the mixture of NTZ and AgNPs showed a significant reduction in parasite numbers after infection [112].

### 3.5. Haemonchus 

Gastrointestinal parasitic nematodes are the most common economic infectious disease in the world [113], with high infection rates and a prevalence in tropical environments, with infants and preschool children being the most vulnerable groups, wherein worm infections can be fatal [114]. *Haemonchus contortus* is a highly infectious parasitic nematode of the gastrointestinal tract that can cause acute anemia, hemorrhagic enteritis, diarrhea, etc. These symptoms can cause a reduction in dairy and meat production by livestock and consequently economic losses [115]. *Haemonchus contortus* is transmitted through the infected soil of various species and can lead to human infection [116]. 

Only a few drugs, such as benzimidazole, imidazothiazole, and ivermectin, have been used in previous treatments for helminths [117]. The prevalence of parasitic infections, the shortage of drugs, and the emergence of drug resistance have made clinical care difficult, and there is an urgent need for a new drug to fill the gap. So, in 2020, some researchers used carvacrol-coated chitosan nanoparticles [118] for antihelminthic activity against the adult stage of *Haemonchus contortus*. Researchers in India have used *Lansium parasiticum* to prepare AgNPs (LAgNPs), as this plant is common in countries such as India and Bangladesh and is mainly used for food and timber purposes. The use of LAgNPs prepared from this plant could open up new avenues for the development of modern medicine as a novel antihelminthic drug. In the study, it was found that 100% of males and 80% of females in samples treated with LAgNPs died after 12 h, but 0% of males and females were paralyzed within one hour in the citric acid-coated AgNPs sample, and only 26% of males and 11.3% of females died after 12 h. Therefore, LAgNPs were more effective against the parasite and had more toxicity. The treatment of *Haemonchus contortus* with LAgNPs showed a faster elevation of ROS and NOS (Nitric Oxide Synthase)-dependent stress in the worms, through which parasite growth was inhibited [75].

### 3.6. Blastocystis Hominis

*Blastocystis hominis* (*B. hominis*) is a unicellular, specialized, anaerobic protozoan that is found mainly in the human gut. It has a prevalence of up to 50% in Egypt [119]. Symptoms of the disease include nausea, vomiting, abdominal cramps, exhaustion, and diarrhea, and are more severe in children and immunocompromised patients. In the Egyptian researchers’ study, AgNPs were used with metronidazole added to their particles [119]. Metronidazole is the drug of choice for the treatment of *Mycobacterium avium*, but since 1976, side effects, treatment failure, and resistance have made the development of a new drug urgent [119]. By comparing AgNPs, AgNPs + MTZ, and MTZ, the drug was introduced into the medium with the parasite in four groups, one of which was a control group without the drug, and the results showed that after three hours, the highest percentage of cyst reduction was 79.67% in the medium where AgNPs + MTZ were placed [119]. Nanoparticles releasing ROS are able to damage the structure of glycoprotein and lipophosphoglycan molecules and also inhibit ATP production and stop DNA replication, so AgNPs can ultimately kill the source of infection and cause the parasite to terminate its infectious activity [120]. Therefore, the combination of AgNPs and conventional drugs seems to the researchers to be a logical direction for future research into the use of AgNPs in the clinic [121].

In a study, also from Egypt, curcumin, a traditional spice in Asian cuisine, was used to prepare the nanoparticles. Quantities of curcumin and poly-D, L-lactide-co-glycolic acid were dissolved in an acetone solution, kept under bath sonication conditions for 2 min, and the emulsion was evaporated for 30 min and the organic solvent removed under magnetic stirring to obtain the nanoparticles. Nanocurcumin can be freely dispersed in water and, in this experiment, nanoparticles. MTZ and untreated controls were used to carry out experiments using three graded concentrations and incubated at 37 °C for 24, 48, and 72 h. The final result was that nanocurcumin at 10 mg/mL showed the highest inhibition rate (92.5%) at 72 h.

### 3.7. Strongylides

*Strongylides* are the main nematode pathogen of horses and are also considered a serious health problem for the herd. Additionally, in clinical treatment, researchers have found the parasite to be resistant to various anthelmintic drugs, particularly benzimidazole, praziquantel, and pyrantel [72]. Brazilian researchers have shown that nematophagous fungi produce extracellular enzymes and AgNPs, which convert toxic metal ions into non-toxic nanoparticles through the combination of enzymes and AgNPs [72]. In previous studies, the insecticidal activity of AgNPs biosynthesized by fungal flagellates against the symbiotic nematode Canis lupus was reported [122]. In the present study, Duddingtonia flagrans AgNPs appeared blue in color and had a diameter of 10 nm, resulting in a 43% reduction in larval numbers compared to the control group, which the researchers attribute to the parasite’s resistance to commercial parasiticides, and could further demonstrate the feasibility of using environmentally friendly D. flagrans fungal AgNPs to control robust nematodes in the future [123].

## 4. Anti-Parasitic Mechanism of AgNPs

AgNPs can be one of the most promising technologies to combat pathogens due to their extensive surface, the release of silver ions (Ag^+^), and the release of ROS, with strong antibacterial and antifungal activity [47]. However, there is a little data on the mechanism of AgNPs for parasitic worms inactivation [109], and the effect of AgNPs on parasites is related to the entry of silver ions and the release of ROS; the released Ag ions can enter the Cryptosporidium oocyst and demolish the sporozoites while nanosized particles of silver can react with the cell wall, resulting in leakage [12]. Metallic silver can disrupt the cell membrane or chemically bind to and deposit on the cell surface, which has a toxic effect on the cell. AgNPs export silver ions, which are toxic to a range of parasites, including protozoan parasites such as *Plasmodium* [124], and the silver ions released can bind to the parasite’s cell membrane and then disrupt the fluidity and integrity of parasite cell membranes, leading to increased permeability and loss of intracellular essential, disrupting normal cell function and ultimately leading to cell death [12].

AgNPs induce the cell apoptosis and destroy parasites mainly through the generation of ROS [125]. Most intracellular stress responses are caused by ROS-mediated toxicity, and oxidative stress is considered the most likely mechanism for AgNP-induced cytotoxicity. The release of silver ions can also cause ROS in the parasite, leading to oxidative stress and damage to cellular components, which can cause the parasite cells to die and eliminate the parasite [126]. A certain amount of ROS is present in the cell in a normal state, and it is maintained in balance with the antioxidant system. After AgNP stress, cells rapidly produce large amounts of ROS, and the antioxidant system expresses a variety of proteins that scavenge excess ROS, such as superoxide dismutase, catalase, glutathione (GSH), thioredoxin, vitamin E, etc. Glutathione can bind and consume ROS; therefore, the glutathione-regulated antioxidant system is considered to be a key defense system for cell survival [127]. AgNPs reduce GSH levels by inhibiting GSH synthase, thus making the cells unable to effectively scavenge intracellular ROS [127]. The imbalance between the production of ROS and their degradation by the antioxidant system can cause oxidative stress, which can lead to many serious adverse effects, such as DNA breakage, mitochondrial damage, peroxidation of proteins and lipids, and ultimately apoptosis [128]. Intracellular ROS overload directs cells to initiate the apoptotic program through p53, protein kinase B (AKT), and mitogen-activated protein kinase (MAPK) signaling pathways. First, AgNP stress causes cells to produce large amounts of ROS, leading to a downregulation of AKT expression, which increases the expression of the pro-apoptotic kinase p38; meanwhile, a decrease in the expression of the DNA repair enzyme PARP leads to a significant increase in the expression of p53, which induces apoptosis (Figure 5) [129].

In addition, silver ions can disrupt ATP synthesis and interfere with the metabolism by breaking the electron transport chain. Silver ions can also interfere with the activity of key enzymes and metabolic pathways in the parasite, leading to cellular dysfunction and death [130].

At this stage, the exact mechanism of action of AgNPs against the parasite is still under investigation, and there may be other effective ways of doing so. However, it is clear that AgNPs have the potential to be a useful tool in the fight against parasitic infections [131].

## 5. Conclusions

The future of AgNPs is promising, with an increasing demand for their unique properties in various applications, and AgNPs are expected to play a significant role in a few areas, such as antimicrobial applications, electronics, energy, medical applications, environmental remediation, etc. This review has summarized recent developments in understanding the antiprotozoal properties of AgNPs. The antiparasitic effect of AgNPs is achieved in different ways, such as by disrupting the parasite’s cell membranes, reducing the ability of metabolic activity, and inducing the parasite to be unable to reproduce properly. AgNPs have potential problems in practice. First of all, in terms of their negative impact on the humans, animals, and environment, if silver is released into the environment, it can become a threat to human health and environmental safety. AgNPs may also have potential effects on animal health, such as a toxic effect on the liver and kidneys. Additionally, the stability of AgNPs is another challenge. Time may weaken the potency, and the reaction with other substances in the environment will change their properties, which may lead to the development of parasite resistance and reduce their clinical antiparasitic efficacy after long-term high use of AgNPs. Therefore, further research and development are needed to solve these problems of the safety, stability, and drug resistance of AgNPs and ensure the safety and sustainable use of materials.

## Figures and Tables

**Figure 1 pharmaceutics-15-01783-f001:**
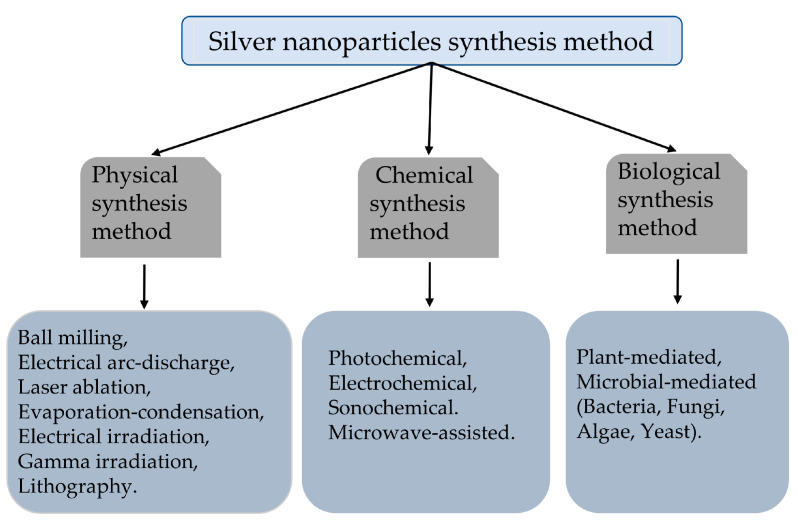
Main synthesis methods of AgNPs.

**Figure 2 pharmaceutics-15-01783-f002:**
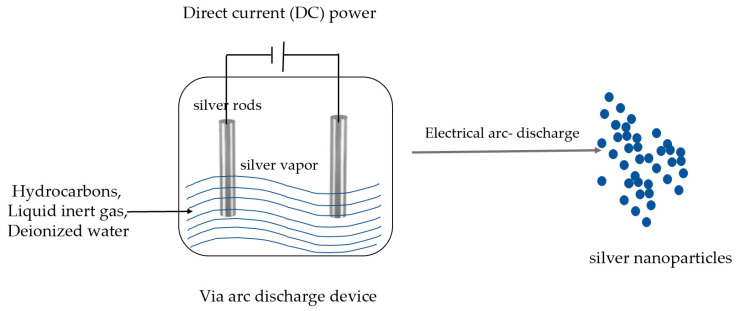
Electrical arc-discharge method of AgNPs.

**Figure 3 pharmaceutics-15-01783-f003:**
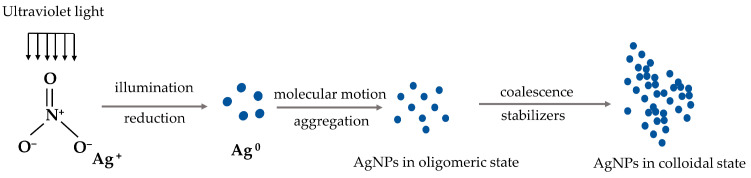
Photochemical method of AgNPs.

**Figure 4 pharmaceutics-15-01783-f004:**
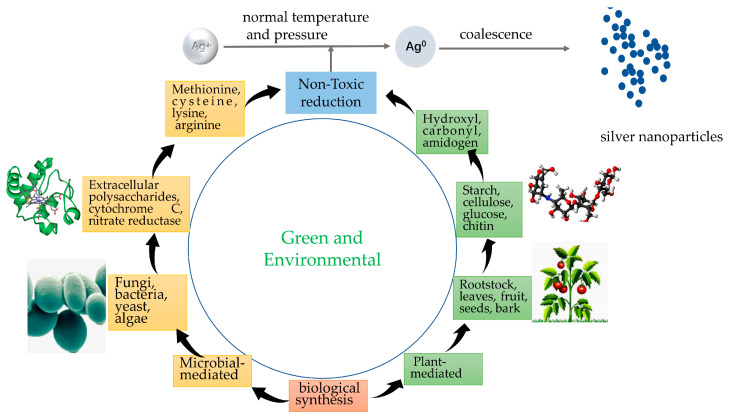
Biological synthesis of AgNPs.

**Figure 5 pharmaceutics-15-01783-f005:**
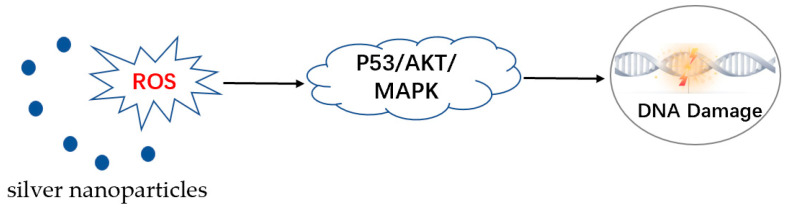
Increased production of ROS causes DNA damage.

## Data Availability

Not applicable.

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
