# Peer review of "Application of Silver Nanoparticles in Parasite Treatment"

_pharmaceutics, 2023, doi:10.3390/pharmaceutics15071783_

Round 1

Reviewer 1 Report

1)    Some minor comments for “Abstract”.

Silver nanoparticles (AgNPs) (are) [is a kind of] ultra-small silver particles with a size (from 1 to 100 nanometers) [between1- 100 nanometers]. Unlike bulk silver, they (have) [can be used in many fields due to their] unique physical and chemical properties [that make them useful in many fields]. [Plenty of] (Numerous) studies have shown that AgNPs (have) [had the] beneficial biological effect(s) on various diseases including antibacterial, anti-inflammatory, antioxidant, antiparasitic, and antiviruses. One of the most well-known (applications) [areas] is in the field of antibacterial applications, (where AgNPs have) [benefited from their] strong ability (abilities) to kill multi-drug resistant bacteria, (making them) [and thus AgNPs is considered as] a potential candidate (as an) antibacterial drug. [In recent years] (Recently), [it has been found that ] AgNPs synthesized from plant extracts (have) exhibits outstanding antiparasitic effects, with shorter duration of use and enhanced ability to inhibit parasite multiplication compared to traditional antiparasitic drugs. This review summarizes the types, characteristics, the mechanism of action in anti-parasitism of AgNPs (in anti-parasitism, focusing on their effects in) [, focusing on its in antiparasitic effect] leishmaniasis, fluke, cryptosporidiosis, toxoplasmosis, Haemonchus, Blastocystis hominis and Strongylides(. The aim is) [, with a view] to provide a reference for the application of AgNPs in the prevention and control of parasitic diseases."

2)    The reviewer suggests that adding a space before all references would improve consistency in formatting. Please check lines 53 (resistance[7]), 57 (forms[7]) , 64 (inhibitors[12]), …

3)    More referencies and information about synthesis methods of AgNPs should be included in figure 1 and line 106. The captions in Figures 1, 2, 3, and 4 are too sumerized, so the need to provide some information about the figures.

Below is some references to add for physical, chemical, and biological synthesis methods in line 106.

Physical methods: This section should include evaporation-condensation, laser ablation, electrical irradiation, gamma irradiation, and lithography.

Lee, D.K. and Kang, Y.S. (2004), Synthesis of Silver Nanocrystallites by a New Thermal Decomposition Method and Their Characterization. ETRI Journal, 26: 252-256. https://doi.org/10.4218/etrij.04.0103.0061

Jae Hee Jung, Hyun Cheol Oh, Hyung Soo Noh, Jun Ho Ji, Sang Soo Kim, Metal nanoparticle generation using a small ceramic heater with a local heating area, Journal of Aerosol Science, Volume 37, Issue 12, 2006, Pages 1662-1670, https://doi.org/10.1016/j.jaerosci.2006.09.002.

Chemical methods:

Mofidfar, M.; Kim, E.S.; Larkin, E.L.; Long, L.; Jennings, W.D.; Ahadian, S.; Ghannoum, M.A.; Wnek, G.E. Antimicrobial Activity of Silver Containing Crosslinked Poly(Acrylic Acid) Fibers. Micromachines 201910, 829. https://doi.org/10.3390/mi10120829

Quintero-Quiroz, C., Acevedo, N., Zapata-Giraldo, J. et al. Optimization of silver nanoparticle synthesis by chemical reduction and evaluation of its antimicrobial and toxic activity. Biomater Res 23, 27 (2019). https://doi.org/10.1186/s40824-019-0173-y

Biological methods:

Mohanta YK, Panda SK, Bastia AK, Mohanta TK. Biosynthesis of Silver Nanoparticles from Protium serratum and Investigation of their Potential Impacts on Food Safety and Control. Frontiers in Microbiology. 2017;8.

Hamouda, R.A., Hussein, M.H., Abo-elmagd, R.A. et al. Synthesis and biological characterization of silver nanoparticles derived from the cyanobacterium Oscillatoria limneticaSci Rep 9, 13071 (2019). https://doi.org/10.1038/s41598-019-49444-y

4)    Can the authors revise experimental trials in vivo or clinical trials in humans for these kinds of AgNPs applications? If there is information, it should be added. If there is not any study,the authors should explain why.

5)    The viewer suggests adding the mentioned references for chemical synthesis to line 122 in order to provide more informtion on the application of redution methods in chemical synthesis methods.  

Ref.: Mofidfar, M.; Kim, E.S.; Larkin, E.L.; Long, L.; Jennings, W.D.; Ahadian, S.; Ghannoum, M.A.; Wnek, G.E. Antimicrobial Activity of Silver Containing Crosslinked Poly(Acrylic Acid) Fibers. Micromachines 201910, 829.

The format for references needs to be consistent. Abstract needs some revisions. 

Reviewer 2 Report

The review article is quite interesting and covers the recent application of silver nano-particles in the treatment of different parasites, however, the mode of action need some inputs and the exact mechanism is still not clear and the author disclosed this limitation in the conclusion of this review., therefore i recommend publishing this review just after addressing the following minor comment:

the following references is advised to be added to section 2; (synthesis of silver nanoparticles):

a) Impact of high throughput green synthesized silver nanoparticles on agronomic traits of onion, International Journal of Biological Macromolecules, 149 (2020) 1304–1317.

b) Utilization of High Throughput Microcrystalline Cellulose Decorated Silver Nanoparticles as an Eco-Nematicide on Root-knot Nematodes, Colloids and Surfaces B: Biointerfaces 188, April (2020) 110805.

c)Eco-friendly method for silver nanoparticles immobilized decorated silica: synthesis & characterization and preliminary Antibacterial activity, Journal of the Taiwan Institute of Chemical Engineers, 95 (2019) 324-331.

The language of this current manuscript should be carefully checked again.

Reviewer 3 Report

The present review describes some aspects of the synthesis of nanosilver particles. It does not refer to a lot of reduction methods of nanosilver obtain like sonication, nuclear irradiation, and ZnO/Ag° (or TiO2/Ag° carriers) nanotechnology, both of them much reported nowadays. 

Some problems, and challenges of the theme are not commented on adequate examples: the biodistribution of the nanosilver, oxidative resistance of Ag°, and efficiency of the nanosilver particles. Why they are not used against so mutable microorganisms? Why the nanosilver geometry is so important, face to penetration in the cell membrane? And so on.

Round 2

Reviewer 1 Report

The reviewer suggests adding some information to the captions of figures. This suggestion is made because captions play an important role in scientific data reporting by helping readers understand and interpret figures accurately.